# OpenReview forum: "PlugGuard: A Streaming Safeguard for Large Models via Latent Dynamics-Guided Risk Detection"
_ICML.cc/2026/Conference — ICML 2026 regular_

### Official Review · Reviewer_wsEF · 2026-03-10

**Soundness:** 3
**Presentation:** 4
**Significance:** 4
**Originality:** 3
**Overall Recommendation:** 4
**Confidence:** 4

**Summary:**

This paper presents PlugGuard, a 20M-parameter plug-in framework designed for real-time, token-by-token risk detection within the Large Model (LM) generation pipeline. The authors seek to move beyond traditional "post-hoc" guardrails—which often allow unsafe content to be exposed before detection—by introducing a streaming moderation system.The proposed method consists of two primary innovations:Streaming Latent Dynamics (SLD): A head that models the temporal evolution of risk across a sequence by leveraging intermediate LM hidden states.Anchored Temporal Consistency (ATC): A training loss that enforces a "benign-then-harmful" temporal prior to stabilize streaming predictions in the absence of fine-grained token labels.The authors also introduce StreamGuardBench, the first benchmark specifically grounded in the native streaming responses of ten widely deployed open-source models. Experimental results show that PlugGuard outperforms state-of-the-art (SOTA) post-hoc guardrails and prior probes by 15.61% in F1 score, while maintaining a sub-millisecond latency overhead ($<0.5$ ms).

**Compliance With Llm Reviewing Policy:**

Affirmed.

**Final Justification:**

The rebuttal is helpful and addresses several of my main concerns, especially regarding the interpretation of the temporal prior and the added SFT comparison. These clarifications improve the paper’s framing and make the empirical evaluation more balanced.

That said, some issues remain, particularly the limited clarification on layer selection and the more modest advantage over strong fine-tuned baselines. Overall, the rebuttal supports my positive view of the paper, but not strongly enough to move beyond weak accept.

**Key Questions For Authors:**

Non-monotonic Risks: How does the ATC loss handle "sandwich" responses where a model provides a refusal, then some harmful content, and finally an ethical disclaimer? Does the monotonicity penalty cause the model to get "stuck" in an unsafe state?

Layer Robustness: Is the "risk-aware" information concentrated in specific layers across all model families (e.g., always the middle layers), or does this vary significantly?

Adversarial Fillers: Could an attacker bypass PlugGuard by padding the beginning of an unsafe response with 100 tokens of "polite filler" to exploit the "head-benign" anchor in the ATC loss?

MT-Bench Trade-off: You report an MT-Bench score of 7.66 for the streaming variant. Can you provide a breakdown of which categories (e.g., Coding, Reasoning) suffered the most from streaming blocking?

**Limitations:**

yes

**Strengths And Weaknesses:**

### Strengths
Practical Significance: The work addresses a critical security principle: ensuring safety before harmful tokens reach the user. This real-time intervention is significantly more practical than withdrawing content after it has already been streamed.

Methodological Innovation: The use of continuous-time latent dynamics (via a discrete gated approximation) to model risk accumulation is a sophisticated departure from simple MLP-based probes.

Empirical Depth: The creation of StreamGuardBench (268k labeled pairs) is a major contribution that addresses the distribution mismatch inherent in static safety datasets.

Operational Efficiency: With only 20M parameters and the ability to run in an asynchronous "parallel mode," the system adds virtually zero perceived latency for the user.

### Weaknesses
Heuristic Nature of the Temporal Prior: The ATC loss relies heavily on the assumption that harmfulness is prefix-monotone (i.e., a response starts benign and turns harmful). While data supports this generally (only 1.25% of samples start harmful), this prior might be fragile against sophisticated adversarial prompts that embed "toxic spikes" or late-stage self-corrections.

Evaluation Fairness: The main text emphasizes a 15.61% F1 improvement, but many baselines (like LlamaGuard3 and ShieldGemma) are evaluated in a zero-shot manner. The appendix reveals that after fine-tuning (SFT), the performance gap narrows significantly, suggesting the SOTA claim should be more carefully qualified.

Layer Selection Ambiguity: PlugGuard requires "tapping" into specific intermediate layers. The paper lacks a systematic analysis or heuristic for how a user should select the optimal layer across different model architectures (e.g., Layer 17 was chosen for Qwen2.5-7B, but no universal rule is provided).

Over-blocking Analysis: While benign prompts are included to assess over-blocking, there is limited analysis of the specific types of safe content that trigger false positives, which is crucial for maintaining model helpfulness.

---

> ### Author Rebuttal · Authors · 2026-03-31
>
> > **1. Heuristic Nature of the Temporal Prior (Q1,W1)**
>
> We would like to clarify that the monotonic temporal prior is not an arbitrary heuristic; it follows directly from the safety goal of streaming moderation.
>
> In our setting, the guardrail evaluates the safety of the generated prefix [1,t] under a strict stop-if-harmful policy. Once harmful content appears, the prefix is already unsafe, and later benign tokens cannot undo that exposure. For example, if a model first provides a suicide method and then adds supportive advice, the response should still be treated as unsafe from the moment the harmful method appears. Therefore, under prefix-level safety, the risk state is naturally monotonic: once unsafe, always unsafe.
>
> This design is also important for resisting “sandwich” attacks such as refusal → harmful content → disclaimer. Once harmful content appears, subsequent disclaimers should not reduce the risk score. This definition is consistent with prior work that treats responses as unsafe as long as they contain harmful content, regardless of later mitigation or disclaimers [A,B].
>
> [A] Yu et al. “GPTFUZZER: Red Teaming Large Language Models with Auto-Generated Jailbreak Prompts.” ArXiv(2023).
>
> [B] Feng et al. “JailbreakLens: Visual Analysis of Jailbreak Attacks Against Large Language Models.” (2024).
>
> > **2. Evaluation Fairness (W2)**
>
> In the main text, we follow prior work by evaluating LlamaGuard3 and ShieldGemma in the zero-shot setting, since both are released as general-purpose safety filters intended for out-of-the-box use.
>
> At the same time, we agree that SFT comparison is a fairer setting. Therefore, as shown in Appendix A.3.2 Table 10, we also fine-tuned these baselines. The results show that PlugGuard (20M) remains highly competitive with much larger SFT guardrails: average F1 over response and streaming levels is 0.8455 for PlugGuard, vs. 0.8394 for ShieldGemma-SFT and 0.7408 for LlamaGuard3-SFT, while maintaining a much better efficiency–accuracy trade-off.
>
> In the revision, we will move the SFT results from the appendix to the main text and clarify our claim: PlugGuard achieves state-of-the-art performance among streaming guardrails, while offering far lower latency (<0.5 ms vs. hundreds of milliseconds). This efficiency–accuracy trade-off is a core contribution of our work.
>
> > **3. Layer Selection Ambiguity (Q2,W3)**:
>
> Please refer to our response to **Reviewer sR4c W2**
>
> > **4. Adversarial Fillers (Q3)**
>
> Thank you for this insightful question. We argue that such a "polite filler" attack would not effectively bypass PlugGuard, for both empirical and modeling reasons.
> - Empirically, this pattern is already common in StreamGuardBench: many jailbreak prompts induce long benign prefixes (e.g., role-play, hypothetical framing) before harmful content appears. In our test set, 31.45% of harmful samples have their first harmful token after 100 tokens, naturally matching this attack scenario. On these late-onset samples, PlugGuard achieves 85.90% streaming recall, compared with 88.24% on early-onset samples (≤100 tokens), a drop of only ~2.3%.
>
> - Mechanistically, PlugGuard is not “locked” into a safe state. The ATC loss only anchors the first few tokens as benign; afterward, it mainly discourages risk decreases, so a long safe prefix followed by a sharp rise is fully compatible with the training objective. In addition, the SLD head can quickly update when harmful cues appear, allowing the risk score to rise rapidly despite earlier benign context.
>
> We also provide a qualitative example in Figure 7, where the response begins with an ethical disclaimer but PlugGuard still detects the later harmful shift and intervenes promptly.
>
> In summary, both our empirical results and model design confirm that PlugGuard remains effective against adversarial fillers. We will include this detailed analysis in the revised manuscript to clarify this robustness.
>
> > **5.Over-blocking Analysis (Q4,W4)**
>
> Thank you for requesting a finer-grained MT-Bench analysis. Our results show that PlugGuard largely preserves helpfulness on objective tasks such as Coding, Math, and Extraction. The small performance drop is mainly concentrated in Writing and Humanities. We find two main types of cases:
>
> - Necessary interceptions. Some MT-Bench prompts evolve into genuinely risky requests. For example, a dialogue asks first about the base rate fallacy in political campaigns, and then requests a detailed election campaign plan based on that. In such cases, interception is aligned with the safety objective.
>
> - Over-blocking cases. We also observe PlugGuard occasionally blocks benign prompts, like drafting a financial report email. This reflects a known safety trade-off: it learns high sensitivity to financial context linked to fraud in training, occasionally over-blocking safe professional contexts.
>
> This case study will be detailed in the revision to demonstrate our trade-off between safety and helpfulness.

---

> > ### Author Rebuttal · Reviewer_wsEF · 2026-04-03
> >
> > The rebuttal addressed several of my main concerns in a helpful way. In particular, the authors clarified that the temporal prior is tied to a prefix-level safety definition under a strict stop-if-harmful policy, which makes the monotonic risk assumption more principled in the streaming setting. The added SFT comparison also improves the fairness of the evaluation and supports a more careful claim centered on the efficiency–accuracy trade-off and performance among streaming guardrails.
> >
> > My remaining concern is mainly about layer selection, which is still not fully clarified in the current rebuttal. In addition, while the added SFT results are useful, the advantage over strong fine-tuned baselines appears more modest than the headline claim in the main text initially suggested. Overall, the rebuttal strengthens the paper and supports my positive assessment, but it does not change my view that the work is best positioned as a weak accept rather than a stronger recommendation. I therefore remain at my original score.

---

> > > ### Author Response · Authors · 2026-04-06
> > >
> > > We deeply appreciate your continued engagement and are encouraged that our clarifications regarding the temporal prior and the SFT evaluation effectively addressed your prior concerns.
> > > We would like to take this final opportunity to provide concrete clarifications on your remaining questions regarding layer selection and the performance gap compared to SFT baselines. We believe these details fully illustrate the practical value of PlugGuard.
> > >
> > > 1. **Clarification on Layer Selection Strategy**
> > >
> > > To resolve any ambiguity, we explicitly describe the deployment strategy used for all experiments.
> > > Initially, we performed a lightweight profiling on the validation set for our primary models, Qwen3-8B (36 layers) and Qwen2.5VL-7B (28 layers). We observed that safety-critical representations consistently peak in the mid-to-late block, specifically at **Layer 20** and **Layer 17** respectively. Crucially, our ablation revealed that the performance variance across adjacent layers in this middle block is surprisingly small (variance < 0.02).
> > > Realizing that exhaustive, per-model hyperparameter searches are unnecessary and to prevent overfitting, we simply adopted these validation-derived indices as our default heuristic for all other models based on their depth class:
> > > - **Shallower Models (24–30 layers)** (e.g., Qwen2.5-Omni-7B): We directly reuse **Layer 17**.
> > > - **Standard LMs (30–40 layers)** (e.g., Llama3.1-8B): We directly reuse **Layer 20**.
> > > - **Deeper Architectures (>40 layers)** (e.g., InternLM3-8B): We default to **Layer 24**.
> > > By applying this simple, unified heuristic, PlugGuard achieves highly competitive performance out-of-the-box. We will formally update the Implementation Details in Section 3.1 to include this explicit rule, ensuring our method is easily reproducible and instantly deployable for new architectures.
> > >
> > > 2. **The Modest SFT Gap vs. The Massive Deployment Advantage**
> > >
> > > We completely agree with your assessment that the F1 advantage over the fine-tuned 8B/9B baselines is modest. However, we would like to respectfully highlight that comparing F1 scores in a vacuum overlooks the fundamental paradigm shift PlugGuard offers for streaming moderation.
> > > To illustrate this, we provide a direct comparison of deploying these methods for token-by-token streaming risk detection:
> > >
> > > |Feature|PlugGuard (Ours)|LlamaGuard3-SFT / ShieldGemma-SFT|
> > > |-|-|-|
> > > |Guardrail Paradigm|**Latent Plug-in Head** (Reuses Base Model's states)|**Standalone LLM** (Independent text-in/text-out model)|
> > > |Active Parameters|**~20 Million**|**8 - 9 Billion** (~400x larger)|
> > > |Streaming Mechanism|Reads cached hidden states at step $t$|Requires passing the full accumulated prefix into a second LLM|
> > > |Per-Token Compute|**1 small SLD projection**|1 Full LLM Forward Pass|
> > > |Streaming Latency|**< 0.5 ms**|**Orders of magnitude higher** (Bound by 8B/9B inference)|
> > > |VRAM Overhead|**< 100 MB**|~**16 GB** (Requires loading a second LLM)|
> > > |Avg. Streaming F1|**0.8455**|0.7408 (LG3) / **0.8394** (SG)|
> > >
> > > While the absolute F1 gap between PlugGuard and ShieldGemma-SFT is indeed small, the operational difference is qualitative. In a real-world streaming scenario, executing a full LLM forward pass for every generated token to check for safety is computationally prohibitive and typically forces systems to fall back on delayed "chunk-level" checks, defeating the purpose of strict real-time intervention.
> > >
> > > In contrast, PlugGuard achieves SOTA-comparable accuracy while making real-time, token-by-token intervention practically deployable. As we committed in our initial rebuttal, we will move the SFT results from the appendix to the main text and precisely clarify our claim: **PlugGuard achieves state-of-the-art performance among streaming guardrails, while offering far lower latency (<0.5 ms vs. orders of magnitude higher for full LLMs).** We believe this efficiency breakthrough is the core contribution of our work.
> > >
> > > We hope this transparent breakdown of our layer heuristic and the efficiency-accuracy trade-off fully resolves your remaining reservations. We are deeply grateful for your rigorous review, which has undoubtedly made our final manuscript much stronger.

---

### Official Review · Reviewer_sR4c · 2026-03-12

**Soundness:** 3
**Presentation:** 3
**Significance:** 3
**Originality:** 3
**Overall Recommendation:** 4
**Confidence:** 3

**Summary:**

PlugGuard is a plug-in framework for detecting harmful content in real time during LM generation. Instead of waiting for the full output and then inspecting it (as conventional guardrails do), PlugGuard monitors the model's intermediate hidden states as tokens are being generated. It has three components: (1) a Streaming Latent Dynamics Head (SLD) that reads hidden states and models how risk evolves over the generation process; (2) an Anchored Temporal Consistency (ATC) loss that enforces a prior that risk scores should increase monotonically for harmful outputs (the "starts benign, becomes harmful" pattern); and (3) StreamGuardBench, a new benchmark for streaming guardrails that evaluates on live model generations rather than static datasets, covering both text and vision-language settings. The paper reports better detection accuracy than existing methods with low latency overhead.

**Compliance With Llm Reviewing Policy:**

Affirmed.

**Key Questions For Authors:**

- Have you tested against known jailbreak methods (GCG, AutoDAN, PAIR)? Can an attacker produce harmful text while keeping hidden-state risk scores low?
- See the weakness for other questions.

**Limitations:**

The limitation is not discussed in the paper. The impact statement is well stated.

**Strengths And Weaknesses:**

**Pros**:
- The problem is essential and meaningful for the real-world deployment. Post-hoc guardrails that wait for the complete output before checking safety leave a window where the user is exposed to harmful content. In chatbot settings, this is a real gap.
- Using internal hidden states rather than generated text is reasonable and intuitively effective.
- StreamGuardBench is a welcome contribution. Evaluating streaming guardrails on actual model outputs (not a fixed dataset) better reflects deployment conditions.
- The plug-and-play design that leaves the protected model untouched is practical, and the reported latency numbers are low.

**Cons**:
- For a safety-critical component, the absence of adversarial robustness evaluation is a notable gap. Can an attacker craft prompts whose hidden states "look" benign to SLD even while producing harmful text?
- There is no ablation on which layer the SLD head taps into. Different layers encode different levels of abstraction, and the best layer probably varies across models and tasks. Without this, it is unclear how much tuning a practitioner would need to do to get PlugGuard working on a new model.
- The comparison is mostly against post-hoc methods, which are not entirely fair since they solve a different problem. Other real-time or streaming moderation approaches (token-level classifiers, systems like the OpenAI Moderation API) would make for a more convincing comparison.

---

> ### Author Rebuttal · Authors · 2026-03-31
>
> > **1. Robustness (Q1, W1)**
>
> Thank you for this important point. We agree that robustness against strong jailbreak attacks is essential. To address this,
>
> - First, StreamGuardBench already includes complex jailbreaks samples. The WildGuard and S-Eval subsets contain high-quality, model-generated jailbreak prompts, including role-play (e.g., "DAN" mode) and nested-scenario attacks (e.g., elaborate fictional settings).
>
> - To further evaluate robustness, we conducted new experiments using PAIR (realistic black-box threat)  on Qwen3-8B. As shown below, PlugGuard maintains strong detection performance against PAIR-generated attacks, outperforming both Qwen3Guard variants.
>
> |F1|LlamaGuard3|ShieldGemma|Qwen3Guard-Gen|Qwen3Guard-Stream|Ours|
> |-|-|-|-|-|-|
> |Response|0.5652|0.8197|0.8621|0.7308|0.8000|
> |Streaming|0.6182|0.8788|0.7547|0.7308|0.8421|
>
>  GCG and AutoDAN require access to the target model's gradients or output token logits. To bypass PlugGuard using GCG, an attacker would need to perform an adaptive white-box attack—optimizing prompts through both the frozen LLM and the SLD head simultaneously to force harmful output while manipulating the hidden states to "look" benign. While theoretically possible, this represents a highly complex threat model beyond standard jailbreaks. We will add a discussion on this vulnerability to adaptive white-box attacks in our new Limitations section.
>
> > **2. Layer Ablation (W2)**
>
> To systematically investigate the impact of different layers, we conducted an ablation across multiple architectures and modalities. The performances of layers at different depths:
>
> |Dataset|Model|Early(~15%)|Mid (~40-50% depth)|Mid-to-Late(~60-75%)|Final (~90-100%)|
> |-|-|-|-|-|-|
> |S-Eval|Qwen3-8B|0.8786|0.9132|**0.9162**|0.9124|
> ||Llama3.1-8B|0.9023|0.9268|0.9168|**0.9299**|
> ||Qwen2.5-Omni|0.8660|0.8841|0.8927|**0.9245**|
> |MMSafety|Qwen2.5-Omni|0.7606|0.8065|**0.8132**|0.8000|
> ||InternVL3-8B|0.7487|0.7755|**0.8367**|0.7767|
>
> As shown in the data, consistent patterns emerge that guide practical deployment:
> - Early layers are usually weaker, likely because they mainly encode low-level lexical/visual features.
> - Final layers are not always optimal, possibly because representations become specialized for next-token prediction.
> - Mid-to-late layers (60–75% depth) provide the most consistent performance across models.
>
> Importantly, the performance variance within the Mid-to-Late region is generally small (often fluctuating by less than 0.02). Thus, practitioners using PlugGuard do not need to perform an exhaustive search for every new model. Selecting a layer at approximately 60-75% of the model depth is a reliable and effective default. In the submission, we used this mid-to-late heuristic without fine-grained tuning. We will add this analysis in revision.
>
> > **3. Streaming Moderation Comparison (W3)**
>
> 1. Streaming Baselines
>
> Thank you for the suggestion. We would like to clarify that our paper already includes comparisons with multiple streaming moderation baselines.
> - Token-Level Guardrails: We explicitly compared PlugGuard against DSA, ShieldHead, SCM, and the recently released Qwen3Guard-Stream (detailed in Tables 1, 11, and 12).
> - Decoding-time Intervention Methods: We compared against active intervention approaches such as SafeDecodingand Model Surgery(detailed in Table 3).
>
> Across all these comprehensive comparisons, PlugGuard consistently demonstrates superior effectiveness and efficiency.
>
> 2. Comparison with OpenAI Moderation API
>
> Besides, we conducted an additional evaluation using the official OpenAI Moderation API. The response-level F1 scores are as follows:
>
> |F1|WildGuard|SEval|FigStep|MMSafety
> |-|-|-|-|-
> |API|0.2461|0.2133|0.3600|0.5067
> |Ours|0.8462|0.9282|0.8089|0.8273
>
> The API performs worse on these benchmarks, but we found this is mainly due to policy/taxonomy mismatch rather than pure capability differences. OpenAI moderation uses a fixed category system (e.g., hate, harassment, self-harm, sexual, violence), while academic benchmarks often include additional risks such as cybersecurity and privacy or use different label definitions. This makes direct comparison with black-box industrial APIs difficult.
> > **4. Limitation**
>
> We sincerely thank the reviewer for the constructive comment. We agree that a clear discussion of limitations is essential for a rigorous scientific contribution. In the revision, we will add a dedicated "Limitations" section:
> - Modality and Language Coverage: As the first benchmark tailored for streaming guardrails, StreamGuardBench currently focuses on text and image-text in English. Future work will extend to audio, video, and multilingual scenarios.
> - Layer Selection Heuristic: While our ablation study shows that PlugGuard is robust when tapping into the "Mid-to-Late" layers (60-75% depth), the optimal layer index is not automatically learned. Future work could explore differentiable layer selection mechanisms to further reduce deployment effort.

---

> > ### Author Rebuttal · Reviewer_sR4c · 2026-04-03
> >
> > Thanks for the detailed response. The authors have addressed my concerns with new experiments and a commitment to add a Limitations section. I maintain my score of 4, as the weaknesses from all other reviewers I identified (e.g., benchmark scale, monolingual focus, lack of token-level annotations) are inherent scope limitations acknowledged as future work rather than fundamental flaws.

---

> > > ### Author Response · Authors · 2026-04-06
> > >
> > > We sincerely thank you for your constructive feedback and the continued engagement throughout the review process.
> > >
> > > We deeply appreciate your insightful observation that the remaining concerns (e.g., benchmark scale, monolingual focus, and lack of token-level annotations) are indeed "inherent scope limitations acknowledged as future work rather than fundamental flaws." We fully agree with this characterization — these scope boundaries (e.g., multilingual extension, multi-turn streaming benchmark) represent natural next steps that do not diminish the contributions of the current work: the first streaming guardrail framework with principled latent dynamics modeling, a new benchmark paradigm, and strong empirical results across 10 models.
> > >
> > > As you noted, our primary goal is to establish the **first** rigorous evaluation standard specifically designed for realistic, on-the-fly streaming moderation. While we entirely agree that expanding to multilingual or audio/video contexts are highly meaningful future extensions, StreamGuardBench currently covers **25 risk categories** across **both text and image-text modalities**, and grounding the evaluation in the actual generation dynamics of **10 representative LLMs and VLMs**. We believe this provides a sufficiently multi-dimensional evaluation to validate the generalizability of streaming guardrails. We will explicitly discuss these scope limitations in the revised "Limitations" section.
> > >
> > > Furthermore, beyond the benchmark, we also introduce a lightweight (20M parameters) plug-in. By leveraging the rich, deep semantic representations of the frozen base model and introducing a novel Anchored Temporal Consistency (ATC) loss with innovative Streaming Latent Dynamics (SLD) Head to model the risk evolution across the generated sequence, our proposed PlugGuard achieves robust, real-time risk detection without the heavy latency overhead of post-hoc models. This provides a highly practical and scalable blueprint for transitioning from reactive post-hoc moderation to proactive, in-decoding safety interventions.
> > >
> > > Given that all your original concerns have been substantively addressed with new experimental evidence and concrete revision commitments, we respectfully hope you might consider whether the current evidence supports an updated assessment. We believe the contributions — a novel and efficient streaming guardrail achieving state-of-the-art among streaming methods with sub-millisecond overhead, plus a new benchmark paradigm — represent a solid advance for the safety community.
> > >
> > > Thank you again for your constructive engagement throughout the review process.

---

### Official Review · Reviewer_aNkH · 2026-03-13

**Soundness:** 3
**Presentation:** 2
**Significance:** 3
**Originality:** 3
**Overall Recommendation:** 2
**Confidence:** 3

**Summary:**

This paper studies streaming safety moderation for large language and vision-language models. The authors argue that existing guardrails are mostly post-hoc and therefore cannot prevent unsafe content from being exposed before detection. To address this, the paper introduces two main contributions. First, it proposes StreamGuardBench, a benchmark for evaluating streaming guardrails using responses generated by the protected models themselves rather than only static external corpora. Second, it presents PlugGuard, a lightweight plug-in safety module that reads intermediate hidden states during decoding and performs token-level risk prediction using a Streaming Latent Dynamics head together with an Anchored Temporal Consistency loss. Experiments on text and multimodal benchmarks show improvements over post-hoc guardrails and prior plug-in methods under low latency overhead.

**Compliance With Llm Reviewing Policy:**

Affirmed.

**Key Questions For Authors:**

1. StreamGuardBench has a relatively limited overall scale, and several subsets, especially the multimodal ones, are quite small. Is the current data scale sufficient to support stable and reproducible evaluation of streaming guardrails? Is there stronger statistical evidence showing that the reported results would remain consistent at a larger scale?

2. The method and loss design appear to rely heavily on a “benign-then-harmful” temporal prior. Is this pattern overrepresented in the benchmark? How does PlugGuard perform on more complex temporal patterns, such as harmful content appearing immediately, alternating between harmful and benign content, or appearing in multiple separate bursts? If such cases are underrepresented, could the current evaluation overestimate the method’s effectiveness?

3. The benchmark mainly provides response-level labels, while the core challenge of streaming guardrails is deciding when to intervene. Without token-level annotations, how does the current evaluation accurately measure whether the intervention timing is appropriate? Is there any finer-grained annotation or stronger evidence supporting the timing-related claims?

4. StreamGuardBench is currently built mostly around English, single-turn, offline, deterministic generation. Why is this setup sufficient to support claims about realistic streaming deployment? What are the applicability limits of the benchmark and the method in multilingual, multi-turn, long-context, non-deterministic generation, and genuinely visual harmful-content settings?

**Limitations:**

The paper discusses limitations to some extent, but I think the discussion should be more explicit about the benchmark’s current weaknesses: limited scale in several subsets, narrow scenario coverage, lack of token-level labels, reliance on model-based annotation, incomplete multimodal coverage, English-centric design, and the mismatch between deterministic offline generation and real deployment conditions.

**Strengths And Weaknesses:**

**Strengths**

The paper addresses an important and practical problem, namely streaming safety moderation for LLMs/VLMs, which is more relevant to real deployment than post-hoc filtering.

A key strength is **StreamGuardBench**, which is among the first benchmarks specifically designed for streaming guardrails and adopts a **model-in-the-loop** construction paradigm, making evaluation closer to target-model generation behavior.

 The benchmark covers both **text-only** and **vision-language** settings, and also provides a relatively unified evaluation protocol, which is valuable for standardized comparison in this emerging area.

**PlugGuard** is lightweight and deployment-oriented, and the reported improvements over post-hoc and prior plug-in baselines under low latency overhead are practically appealing.

**Weakness**

1.Scale and Coverage Deficiencies: The dataset's overall scale (~268k pairs) and its subset distributions (e.g., only 200 samples for multi-modal evaluation) are insufficient to support high-confidence benchmarking. It predominantly focuses on conventional, direct attacks, lacking adequate coverage of complex, long-tail adversarial scenarios (e.g., financial fraud, subtle manipulation of minors).

2.Unrealistic Temporal Prior: The temporal distribution of toxic tokens is highly skewed and fails to reflect real-world streaming dynamics. With only 1.25% of toxic content appearing within the first 10 tokens, the dataset follows a predictable "benign-then-toxic" pattern. This structural bias is highly likely to induce shortcut learning in the evaluated models, undermining the validity of the benchmark.

3.Static and Monolingual Constraints: The benchmark relies entirely on offline-generated, single-turn English interactions. It completely ignores critical real-world streaming variables, such as multi-turn progressive jailbreaks, dynamic decoding strategies (e.g., temperature/top-p sampling), user interruptions, and complex multilingual/cross-lingual contexts.

4.Superficial Multi-modal Evaluation: The dataset's multi-modal claim is fundamentally flawed. It only covers simple "malicious text embedded in images" (via OCR) and completely fails to evaluate natively toxic visual content (e.g., violent/NSFW imagery), not to mention the absence of audio/video modalities essential for modern streaming guardrails.

5.Lack of Token-Level Annotations: As a benchmark explicitly designed for streaming scenarios, providing only coarse-grained (sentence/paragraph-level) binary labels is a fatal flaw. The absence of token-level boundary annotations means the benchmark can only evaluate static interception accuracy, failing to quantify the most critical metric for streaming guardrails: interception latency/timing (i.e., exactly when the model triggers the block).

6.Cascaded Errors in Multi-modal Ground Truth: The reliance on an "OCR + Text LLM" pipeline for annotating multi-modal samples introduces systematic noise. This pipeline cannot detect natively toxic visual elements, and OCR inaccuracies will inevitably cascade, severely compromising the reliability and validity of the ground truth.

---

> ### Author Rebuttal · Authors · 2026-03-31
>
> > **1. Scale, Coverage & Multimodal Evaluation (W1, W4, W6, Q1)**
>
> 1. Scale & Coverage: StreamGuardBench is one of the largest safety benchmarks available, larger than widely used datasets like WildGuard (39k), ToxicChat (10k), and comparable to BeaverTails (330k) (Table 7) and covers long-tail risks (financial fraud, social ethics, etc.). Cross-model consistency (Table 5) and generalization on BeaverTails (330k) (Table 12) provide strong evidence that the current data scale supports robust and reproducible evaluation.
>
> 2. Multi-Modal Benchmark
> - Why: We selected datasets based on strict criteria: Scale(>1k), Risk Coverage, Multi-modal Interaction (both visual and textual inputs contribute to the harmful intent), and capability of inducing unsafe responses. Thus, we specifically selected the mmsafety and figstep.
> - Toxic visual/audio content: We also expanded our benchmark to include SafeBench (13.8k diverse samples of native image/audio/text harms) below. PlugGuard strongly outperformed the post-hoc OmniGuard baseline and streaming DSA.
>
> |Dataset|OmniGuard||DSA||PlugGuard||
> |-|-|-|-|-|-|-|
> ||response|streaming|response|streaming|response|streaming
> |SafeBench|0.4797|0.4713|0.7881|0.5171|0.8654|0.7845
>
> - OCR annotation: Our initial multi-modal set targeted typographic attacks, where text is deliberately embedded in images without toxic visual content. For this specific threat, the OCR+LLM pipeline captures the exact malicious payload without information loss, and our human-in-the-loop audit minimizes cascade noise (Appendix A.1).
>
> > **2. Unrealistic Temporal Prior (W2, Q2)**
>
> Due to strict character limits, please refer to our response to Reviewer wsEF (Q1,W1) for a detailed discussion on this.
>
> > **3. Static and Monolingual Constraints (W3, Q4, Q5)**
>
> - Static: We utilized deterministic decoding to ensure strict reproducibility and fair comparison across different guardrails. To validate this, we evaluated PlugGuard and other streaming guardrails on the WildGuard test set using dynamic decoding (temperature=0.7, top_p=0.9, top_k=20) over 5 run. The results confirm that: (1) Randomness introduces minor variance but does not alter the relative performance ranking.  (2) PlugGuard maintains high performance with low variance.
>
> ||DSA|ShieldHead|PlugGuard
> |-|-|-|-|
> |Deterministic/Random|0.69/0.67±0.01|0.58/0.54±0.04|0.85/0.83±0.02
>
> - Monolingual: Similar to prior works in this streaming guardrails (e.g., DSA, ShieldHead), StreamGuardBench currently focuses on English to align with mainstream safety datasets. While expanding to diverse multilingual contexts is the next frontier, to address your concern, we conducted a new zero-shot cross-lingual evaluation on 7 languages (covering both high/low resource) with our English-trained PlugGuard. Despite using only 20M parameters, PlugGuard demonstrates strong zero-shot cross-lingual transferability, validating that it learns language-agnostic risk representations.
>
> ||en|zh|it|vi|ar|ko|th|jv
> |-|-|-|-|-|-|-|-|-
> |DSA|0.69|0.57|0.57|0.50|0.57|0.49|0.43|0.40
> |ShieldHead|0.59|0.40|0.15|0.14|0.26|0.34|0.19|0.25
> |PlugGuard|**0.85**|**0.72**|**0.80**|**0.78**|**0.79**|**0.79**|**0.76**|**0.72**
>
> - Mutil-turn and Complex Adversarial Scenarios: Constructing a statistically significant multi-turn streaming benchmark requires a highly complex, closed-loop dynamic pipeline (where adversarial turns iteratively adapt to real-time model rejections). While this constitutes an entire research scope for our next version, we evaluated PlugGuard against PAIR-based adversarial attacks (which simulate sophisticated single-turn jailbreaks) as a proxy to validate robustness against complex adversarial intent. As shown in the PAIR evaluation table (the first table) in our response for Reviewer sR4c, our method successfully identifies these highly evasive attack trajectories, proving its applicability to complex, progressive malicious intents.
>
> > **4.Token-Level Annotations (W5, Q3)**
>
> Existing static benchmarks cannot answer if a plug-in guardrail will intercept a specific target model's unique trajectory. StreamGuardBench solves this prerequisite "model-in-the-loop" evaluation challenge. We prioritized response-level labels for scalability, as exact token-level risk boundaries are highly subjective and costly to annotate.
>
> To validate timing, we augmented the WildGuard test set with token-level boundaries using LLM-assisted human verification. PlugGuard revealed a Median Leaked Tokens of 0.00 and a Proactive Intervention Rate of 64.06% (delay ≤ 10 tokens). For complex jailbreaks, PlugGuard capped Average Leaked Tokens at 62.85, whereas traditional post-hoc guardrails leaked the entire response (>800 tokens), yielding a >92% reduction in user exposure.
>
> > **5. Summary**
>
> We will explicitly add a **Limitations and Broader Applicability** section in the revised manuscript incorporating these in-depth discussions (scale, scenario coverage, annotation subjectivity) and the new experimental results.

---

### Decision · Program_Chairs · 2026-04-30

**Decision:**

Accept (regular)

**Comment:**

This paper proposes a streaming safeguard for LLMs and VLMs. Rather than detecting risks after generation, the method performs streaming risk detection during generation based on intermediate hidden states. It also introduces StreamGuardBench, a new benchmark featuring on-the-fly responses from each protected model.

Reviewers highlight several strengths. The paper addresses an important and practical problem in streaming safety moderation. StreamGuardBench is the first benchmark specifically designed for realistic streaming moderation. The proposed method, PlugGuard, is a lightweight plug-in with only 20M parameters.

Remaining concerns include:

- The advantage over strong fine-tuned baselines is marginal. The newly added cost and latency comparison is helpful, but a more comprehensive comparison and discussion are needed in the revised version if efficiency is positioned as the primary benefit.
- Some algorithm design choices are not well justified. For example, the layer selection strategy lacks a clear explanation, and there is no systematic analysis for selecting the optimal layer across different model architectures.